# Effects of Canine-Obtained Lactic-Acid Bacteria on the Fecal Microbiota and Inflammatory Markers in Dogs Receiving Non-Steroidal Anti-Inflammatory Treatment

**DOI:** 10.3390/ani12192519

**Published:** 2022-09-21

**Authors:** Kristin M. V. Herstad, Hilde Vinje, Ellen Skancke, Terese Næverdal, Francisca Corral, Ann-Katrin Llarena, Romy M. Heilmann, Jan S. Suchodolski, Joerg M. Steiner, Nicole Frost Nyquist

**Affiliations:** 1Department of Companion Animal Clinical Sciences, Faculty of Veterinary Medicine, The Norwegian University of Life Sciences, 1433 Aas, Norway; 2Faculty of Chemistry, Biotechnology and Food Sciences, The Norwegian University of Life Sciences, 1433 Aas, Norway; 3EMPET Skedsmo Dyresykehus, 2007 Kjeller, Norway; 4Department of Paraclinical Sciences, Faculty of Veterinary Medicine, The Norwegian University of Life Sciences, 1433 Aas, Norway; 5Department for Small Animals, Veterinary Teaching Hospital, College of Veterinary Medicine, University of Leipzig, DE-04103 Leipzig, SN, Germany; 6Gastrointestinal Laboratory, Department of Small Animal Clinical Sciences, College of Veterinary Medicine and Biomedical Sciences, Texas A&M University, 4474 TAMU, College Station, TX 77843, USA

**Keywords:** probiotics, NSAID-induced enteropathy, intestinal dysbiosis, inflammatory biomarkers

## Abstract

**Simple Summary:**

The use of non-steroidal anti-inflammatory drugs (NSAIDs) has prolonged the longevity and well-being of dogs with osteoarthritis and other painful conditions. However, this treatment is also associated with diarrhea in dogs, but the pathogenetic mechanisms and possible prevention strategies remain unknown. This study aimed to determine whether canine-obtained lactic acid bacteria affect the frequency of diarrhea, fecal microbiota (dysbiosis index), and gastrointestinal inflammation (assessed by calprotectin and S100A12/Calgranulin C) in dogs receiving NSAIDs. Diarrhea occurred in 4/12 dogs (33%) receiving placebo and 1/10 dogs (10%) receiving canine-obtained lactic acid bacteria (LAB), but this difference was not significant. The fecal dysbiosis index, calprotectin, and S100A12 were not significantly different between dogs receiving NSAIDs and LAB and dogs receiving NSAIDs and placebo. This study suggests that LAB is safe to use in NSAID-treated dogs, but further studies are needed to determine its potential to ameliorate diarrhea and gastrointestinal inflammation in dogs receiving NSAIDs.

**Abstract:**

Non-steroidal anti-inflammatory drugs (NSAIDs) may cause enteropathy in dogs and probiotics may be one option to prevent this. The objective of this study was to determine whether the administration of canine-obtained lactic acid bacteria (LAB) has an effect on the frequency of diarrhea, the composition of the fecal microbiota, and/or markers of gastrointestinal inflammation in dogs receiving NSAIDs when compared to dogs given NSAIDs and a placebo. A total of 22 dogs treated with NSAIDs for various clinical indications were enrolled in a seven-day randomized, double-blinded placebo-controlled interventional study. Dogs were randomized to receive either placebo or LAB, a product containing *Limosilactobacillus fermentum*, *Lacticaseibacillus rhamnosus*, and *Lactiplantibacillus plantarum*. Fecal samples were collected on days one and seven. The fecal microbiota was evaluated using the fecal dysbiosis index (DI) and individual bacterial taxa. Fecal calprotectin (CP) and S100A12/Calgranulin C concentrations were used as markers of gastrointestinal inflammation. There was a difference in frequency of diarrhea between groups, with it affecting 4/12 dogs (33%) in the placebo group and 1/10 dogs (10%) in the LAB group, but this difference did not reach statistical significance (*p* = 0.32). There was a correlation between S100A12 and CP (*p* < 0.001), and *Clostridium perfringens* correlated with S100A12 (*p* < 0.015). Neither treatment significantly affected S100A12 (*p* = 0.37), CP (*p* = 0.12), or fecal DI (*p* = 0.65). This study suggests that LAB is a safe supplement to use for short-term treatment in NSAID-treated dogs, but further studies are needed to determine its potential to prevent NSAID-induced enteropathy in dogs.

## 1. Introduction

Non-steroidal anti-inflammatory drugs (NSAIDs) are the most commonly prescribed analgesics in veterinary medicine [1]. The introduction of NSAIDs has prolonged the longevity and well-being of dogs with osteoarthritis and other painful conditions [2]. However, treatment with NSAIDs may have side effects, most commonly involving the gastrointestinal (GI) tract [3,4]. NSAID-induced gastric ulcerations are related to the decreased perfusion of the gastric mucosa due to a lack of prostaglandins, followed by damage through the actions of gastric acid [4]. However, NSAID-induced lesions may also occur in the lower part of the GI tract where other mechanisms are involved. Indeed, a study using video capsule endoscopy documented that the majority (10/12, 83%) of dogs receiving long-term NSAID treatment had GI lesions involving the distal part of the small intestine [5]. The mechanisms behind these GI lesions include alterations in the adherence and mucosal invasion of intestinal microbes and intestinal dysbiosis [6]. If dysbiosis plays a role in NSAID-induced GI lesions, the modulation of the intestinal microbiota may reduce such side effects [6,7]. 

Several studies suggest that the intestinal microbiota may have an impact on NSAID-induced side effects in the GI tract [8,9,10]. For example, germ-free animals are resistant to NSAID-induced enteropathy. Facultative anaerobic bacteria, such as *Escherichia coli*, *Klebsiella* spp., and *Proteus* spp., identified in rats were reported to contribute to ulcer formation, whereas *Lactobacillus* spp. and *Bifidobacterium* spp. were found to prevent their development, possibly by repressing the establishment of ulcer-associated bacteria [11]. The degree of intestinal dysbiosis can be evaluated using the fecal dysbiosis index (DI), which is a mathematical algorithm based on the results of quantitative PCR (qPCR) assays including seven key bacterial taxa (*Faecalibacterium* spp., *Turicibacter* spp., *Streptococcus* spp., *E. coli*, *Blautia* spp., *Fusobacterium* spp., and *Clostridium hiranonis*) [12]. 

In dogs with chronic enteropathy, biomarkers suggestive of GI inflammation include fecal calgranulin C (S100A12) and calprotectin (CP) [13]. In dogs with chronic inflammatory enteropathy, fecal S100A12 and CP have been shown to correlate with clinical disease [14,15]. Probiotics are live microorganisms consumed orally to promote a healthy gut state [16]. Studies in humans and laboratory rodents have shown that administration of probiotics together with NSAIDs is seen as one option to reduce the risk of NSAID-induced gastroenteropathy [17,18,19], although one study in rats did not find favorable effects of combining NSAIDS with probiotics [20]. To the best of the authors’ knowledge, no previous studies have evaluated the effects of lactic acid bacteria or probiotics in dogs treated with NSAIDs. 

In this study, we aimed to evaluate the frequency of diarrhea, markers of GI inflammation (fecal CP and S100A12) and DI in NSAID-treated dogs administered LAB compared to those administered a placebo. We hypothesized that NSAID-treated dogs receiving LAB would have less diarrhea, lower fecal CP and S100A12 concentrations, and a lower fecal DI. 

## 2. Materials and Methods

### 2.1. Animals

The study protocol was reviewed and approved according to the ethics committee guidelines at the Faculty of Veterinary Medicine, Norwegian University of Life Sciences (NMBU) (approval number: 14/04723-63). Written informed consent was obtained from all dog owners before participation, and they were informed that their participation in the study was voluntary. 

Client-owned dogs initiating NSAID treatment regardless of the indication for treatment were included in the study. The type of NSAID used was at the veterinarian’s discretion. However, dogs were not included if NSAIDs had been administered within the last three months prior to inclusion in the study. The dogs were fed their usual diets consisting of various commercial dry foods and were not restricted to any specific diet throughout the study trial. Any episodes of hyporexia/anorexia were reported. One dog in the LAB group and one dog in the placebo group received antibiotics (amoxicillin) as part of the treatment plan to manage their conditions. None of the dogs received proton pump inhibitors (PPIs). 

### 2.2. Lactic Acid Bacillus (LAB) and Placebo Products

The LAB was designed to contain lactic acid bacteria that had been cultured from healthy dogs [21] and consisted of *Limosilactobacillus fermentum*, *Lacticaseibacillus rhamnosus*, and *Lactiplantibacillus plantarum* fermented in milk. The placebo product was powdered micro-crystallized cellulose. Dog owners were instructed to administer 1 teaspoon (∼5 g) LAB/placebo (for dogs weighing > 3 kg) or ½ teaspoon (∼2.5 g) LAB/placebo (for dogs weighing < 3 kg) once daily, which for LAB corresponded to 6.2 × 108 and 3.1 × 108 colony-forming units (CFU), respectively, of each of the bacteria. The powder was either sprinkled over the food or diluted in water and given orally by syringe.

### 2.3. Study Design

The study was a seven-day randomized double-blinded placebo-controlled interventional trial (Figure 1). Dog owners were instructed to record their dog’s appetite, fecal consistency, and any episodes of emesis daily during the seven-day trial period. Fecal consistency was recorded as either watery, loose, normal, or hard. The term “diarrhea” was used if the fecal quality was watery. All dogs were randomized to receive either LAB or placebo for seven days. The randomization process was performed by block randomization using a block size of six (random.org). In dogs that developed GI side effects, NSAID treatment was discontinued if deemed necessary by the attending veterinarian. 

Fecal samples were collected on the first day of the study (day, D1) and at the end of the study (day, D7). When diarrhea required the discontinuation of NSAID treatment, the second fecal sample was collected on the last day of treatment. Fecal samples were collected immediately following natural defecation. The sample was further divided into two aliquots deposited into sterile plastic containers and frozen immediately, either in the owner’s home freezer (−20 °C) and then transported on dry ice for storage at −80 °C at the central storage unit, or frozen immediately at −80 °C. Fecal samples were sent to the Gastrointestinal Laboratory at Texas A&M University (TAMU) on dry ice to measure fecal CP and S100A12 concentrations and determine the fecal DI.

### 2.4. Microbiota Analyses

The fecal microbiota was evaluated based on the fecal DI [12] using quantitative PCR, as described previously [22]. DI > 2 indicates intestinal dysbiosis and values between 0 and 2 were considered equivocal. We also quantified fecal abundances of *Clostridium perfringens* [23] and *Lactobacillus* spp. [24] using qPCR. Briefly, fecal DNA was extracted using the QIAmp PowerFecal Pro DNA KIT (Qiagen) and an automatic extraction system (Thermo KingFisher Flex Magnetic Particle Purification 96 PCR Isolation system), following the manufacturers’ instructions. The qPCR assays were performed using a Bio-Rad C1000 Touch Thermal Cycler (Bio-Rad Laboratories, California, USA) with the following protocol: initial denaturation at 98 °C for 2 min; 35 cycles with denaturation at 98 °C for 3 s; and annealing for 3 s. All samples were run in duplicates and the average of the two samples was used for further analyses. The qPCR results were analyzed using the Bio-Rad CFX Maestro 1.1 software (Bio-Rad Laboratories). The qPCR data for the individual bacterial taxa (*Faecalibacterium* spp.; *Turicibacter* spp.; *Streptococcus* spp.; *E.coli*, *Blautia* spp.; *Fusobacterium* spp., *Clostridium hiranonis*, *Clostridium perfringens*, and *Lactobacillus* spp.) were normalized to the qPCR data for total bacteria [22]. 

### 2.5. Markers of Gastrointestinal Inflammation

Fecal CP concentrations were measured by a fully analytically validated species-specific sandwich ELISA, as described previously [15,25], and reported as ng/g [26] with the current reference interval (RI) used at the Gastrointestinal Laboratory at Texas A&M University, TX, USA (0–961 ng/g). Fecal S100A12 concentrations were measured using a fully analytically validated species-specific in-house sandwich ELISA with an RI of 2–484 ng/g [27,28].

### 2.6. Statistical Analyses

Statistical analyses were performed using Prism v8, GraphPad Software Inc, San Diego, CA, USA, and R software v. 2021.9.1.372 (RStudio Team (2021). Rstudio: Integrated Development for R. Rstudio, PBC, Boston, MA, USA URL http://www.rstudio.com/, accessed on 26 March 2021). Due to unequal variances between the groups, a Welch’s t-test was used to test for significant differences from D1 to D7 for S100A12, CP, individual bacterial taxa, and DI between the LAB group and the placebo group. Pearson’s product-moment correlation tested the correlation between inflammatory markers and bacterial taxa. Fisher’s exact test was used to test for differences in the frequency of diarrhea between the LAB and placebo groups during the study period. A principal component analysis (PCA) was conducted on the differences between D1 and D7 for fecal DI, bacterial taxa, fecal CP, and fecal S100A12, using the functions “prcomp” and “autoplot” in the ggfortify package in R. Statistical significance for all tests was set at *p* < 0.05. 

## 3. Results

### 3.1. Demographic and Clinical Factors

A total of 22 dogs were enrolled in the study, of which 10 dogs received LAB and 12 received placebo. The study population consisted of dogs of various breeds, both sexes, and different ages. Dogs in the LAB group were between 4 months and 14 years of age with a median of 6 years, while dogs in the placebo group were between 1 and 10 years of age with a median of 5.9 years (Table 1). The dogs received various commercial diets and one dog in each group received an antibiotic (amoxicillin). However, as detected by Cook’s distance, this did not influence the final result. 

Robenacoxib was used in 7 out of 10 dogs (70%) in the LAB group and 8 out of 12 dogs (66%) in the placebo group, whereas meloxicam was given to 2 out of 10 (20%) in the LAB group and 3 out of 12 (25%) in the placebo group. The type of NSAID being administered was not recorded for one dog in the LAB group and one dog in the placebo group. The reason for NSAID prescription was an orthopedic condition in 3/10 dogs (30%) in the LAB group and 3/12 dogs (25%) in the placebo group. A surgical procedure under general anesthesia had been performed in 8/10 dogs (80%) in the LAB group and 10/12 dogs (83%) in the placebo group. Of these dogs, diarrhea occurred in 1/8 (12.5%) in the LAB group and 3/10 (30%) in the placebo group. 

Of the dogs receiving placebo, 4/12 dogs (33%) developed diarrhea, while this only occurred in 1/10 dogs (10%) in the LAB group. However, this difference did not reach statistical significance (Fisher’s exact test, odds ratio: 0.24, *p* = 0.32). The NSAID treatment was discontinued because of the severity of diarrhea in one of the dogs receiving LAB and two of the dogs receiving placebo. Two dogs receiving LAB vomited on D2 and D7, respectively, whereas one dog receiving placebo vomited on D7 after initiating NSAID treatment. The dogs had normal appetite throughout the study. 

### 3.2. PCA Analyses

The PCA analysis on the change from D1 to D7 revealed no clear distinction between dogs receiving LAB vs. placebo, but indicated that changes in fecal CP and S100A12 concentrations between D1 and D7 in the individual dogs are strongly associated with the abundances of *C. perfringens*. These span the principal component 2 (PC2), which explains 22.3% of the variation in our dataset. The bacterial taxa, except for *E. coli*, can be interpreted as in contrast with the fecal DI, and span the PC1, explaining 29.3% of the variation in the data (Figure 2). The correlation between fecal S100A12 and CP was significant (Pearson’s correlation coefficient = 0.63, *p* < 0.001). The abundance of *C. perfringens* correlated significantly with S100A12 (Pearson’s correlation coefficient = 0.510, *p* = 0.015), but not with CP (Pearson’s correlation coefficient = 0.379, *p* = 0.084).

### 3.3. CP and S100A12 Concentrations, DI and Bacterial Taxa in Dogs Receiving LAB vs. Placebo

The CP concentration in the three dogs with the highest levels at D1 in the LAB group reduced their levels to below 50 ng/g at D7. The same three dogs also had the highest S100A12 concentration at D1, which dropped to a level below 20 ng/g at D7. In the placebo group, five dogs had a CP concentration above the RI (0–961 ng/g) at D7 and the same five dogs had the highest S100A12 concentrations at D7 (Figure 3). There were no significant differences in the levels of S100A12 (*p* = 0.37) or CP (*p* = 0.12) between dogs receiving LAB vs. placebo.

Neither LAB nor placebo had any significant effect on DI (*p* = 0.65) or any of the bacterial taxa during the study period (all *p* > 0.05) (Figure 4).

## 4. Discussion

As far as the authors are aware, no previous studies have investigated the potential protective effects of orally administered canine-obtained lactic acid bacteria or probiotics in dogs given NSAIDs. Among the dogs given the placebo, four dogs (33%) developed diarrhea compared to only one dog (10%) in the LAB group. This difference was not significant and indicates that LAB is safe to use in dogs. However, larger studies are needed to determine whether LAB can prevent diarrhea in dogs given NSAIDs. We found that S100A12 and CP were strongly correlated, as was also demonstrated in a previous study of dogs with chronic inflammatory enteropathy [29]. Five dogs given placebo had increased CP concentrations above the upper limit of the RI at D7, whereas all dogs given LAB had negligible CP concentrations at D7. Fecal CP has been shown to be a sensitive screening marker for NSAID-induced enteropathy in human patients [30], even with short-term treatment (seven days) [31]. Moreover, a study showed that humans treated with NSAIDs and probiotics had decreased fecal CP concentrations compared to those given NSAIDs and placebo [32]. However, another study did not find a beneficial effect of probiotics in humans given NSAIDs [33]. 

As for CP, S100A12 has also been shown to be increased in dogs with chronic inflammatory enteropathy [29], and has been used in humans to separate patients with inflammatory bowel disease from those with irritable bowel disease [34]. S100A12 may therefore potentially be useful as screening marker for NSAID-induced enteropathy. We found that the changes in S100A12 concentrations and *C. perfringens* abundance between D1 and D7 were significantly correlated, suggesting that this bacterial taxon may play a role in NSAID-induced enteropathy. Its role in inflammation is not a new phenomenon, as *C. perfringens* is associated with acute hemorrhagic diarrhea in dogs [35,36,37], where its pathogenetic potential is linked to the production of *netE* and *netF* toxins [38]. 

Previous studies have found that the DI can change in response to diet [39,40], and it can be useful for differentiating dogs with chronic enteropathy from healthy dogs [12]. A previous study of dogs with diarrhea found that the LAB product used could resolve diarrhea and reduce fecal abundances of *C. perfringens* and *Enterococcus faecium* [41], indicating that LAB may potentially cause changes in the bacterial populations and improve gut health. Although we did not find any significant change in DI between the groups, there might be changes in the microbiota composition that would be detected using high-throughput sequencing methods and, ideally, DNA shotgun sequencing. 

There was no significant change in the fecal abundance of *C. hiranonis*, a key bacterium in bile acid metabolism, in dogs receiving LAB vs. placebo. Interestingly, studies in rodents and cell culture systems have found that the enterohepatic circulation of NSAIDs is associated with higher levels of secondary bile acids, which can damage intestinal cells [42,43]. However, whether the dogs in this study also had changes in absolute fecal primary and secondary bile acid concentrations was not determined. 

Dogs receiving LAB in our study did not have a higher fecal abundances of *Lactobacillus* than the dogs in the placebo group. Previous studies have demonstrated higher numbers of these bacteria in fecal samples from dogs given LAB, and the modulated intestinal microbiota was characterized by an increased number of other variants of lactic-acid bacteria [44]. It is possible that NSAID-induced changes in the composition of the microbiota take longer to develop, albeit short-term NSAID use was found to change the intestinal microbiota in humans and rats [45]. Furthermore, in our study, dogs were initiated on LAB and NSAID simultaneously. It is possible that the LAB product would have had a more pronounced effect if given prior to initiating NSAID treatment. A recent study demonstrated that mice given oral probiotics containing lactic acid bacteria five days before NSAID treatment showed enrichment of colonic anaerobes and Lactobacilli, whereas the total abundance of Enterobacter decreased and ameliorated GI inflammation was detected compared to the controls [19]. 

We cannot exclude a confounding effect of general anesthesia, surgery, and/or other medications contributing to the development of GI signs, as we could not include a control group of dogs not receiving NSAIDs due to ethical constraints. However, no differences were detected between the groups of dogs with diarrhea that underwent a surgical procedure (1/3 dogs in the LAB group and 3/9 dogs in the placebo group). Thus, NSAID treatment may exacerbate GI signs and potential GI lesions in these dogs, regardless of the initiating factors. The dogs in our study were fed different diets; thus, we cannot rule out the influence of diet on the dogs’ microbiota composition. However, all dogs were fed commercial dry food diets, no dogs had a change of diet during the trial period, and no dogs ate raw meat or home-cooked diets which may influence microbiota composition [46,47]. 

NSAID-induced enteropathy may not be associated with clinical signs, and diarrhea may not be a precise indicator for this condition [5]. In humans, clinical signs of NSAID-induced enteropathy are nonspecific and, in addition to diarrhea, may include signs of iron-deficiency anemia, GI protein loss, indigestion, constipation, and abdominal pain [6], but affected patients can also be asymptomatic [48]. Therefore, further studies are needed to determine how NSAID-induced enteropathy manifests in dogs. For this purpose, capsule endoscopy can be used to identify gastrointestinal lesions in NSAID-treated dogs. These findings should be correlated with clinical signs such as diarrhea, vomiting, anorexia/hyporexia and weight loss, and inflammatory markers (calprotectin and S100A12). It is also unknown which type of probiotic bacteria and at what dose would be most beneficial to use [7]. 

Notably, as this work was based on a small number of dogs, a larger study is required to document the effect of lactic acid bacteria on GI health in dogs. 

## 5. Conclusions

This study did not find a significant difference in the frequency of diarrhea or change in the DI or individual bacteria taxa in NSAID-treated dogs given LAB vs. placebo. Further studies are needed to evaluate the potential of lactic-acid bacteria to ameliorate adverse GI effects induced by NSAIDs.

## Figures and Tables

**Figure 1 animals-12-02519-f001:**
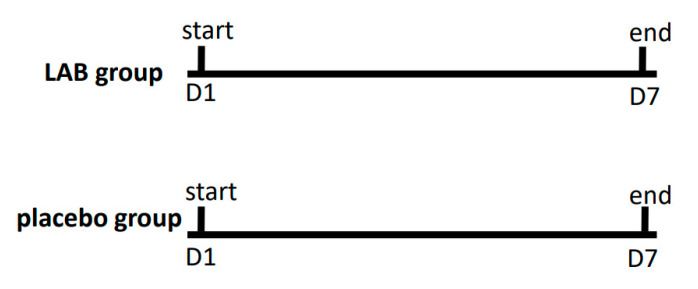
Overview of the study design. Dogs were randomized to receive either LAB or placebo during a seven-day interventional trial. Fecal samples were obtained on days one (D1) and seven (D7) of NSAID plus LAB or NSAID plus placebo treatment. The second fecal sample was taken on the last day of NSAID treatment in dogs that discontinued the study due to developing diarrhea. Fecal consistency was recorded daily throughout the study.

**Figure 2 animals-12-02519-f002:**
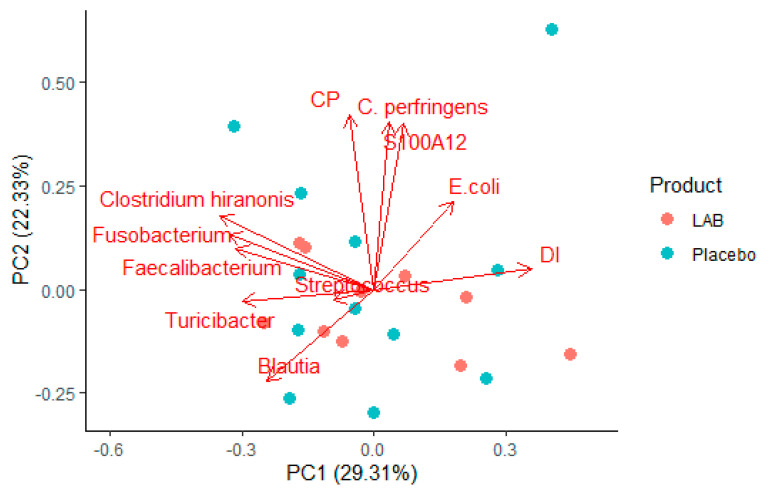
PCA plot showing the changes in the different variables from D1 to D7 in individual dogs given LAB vs. placebo. Fecal CP and S100A12 are strongly correlated and span out on the principal component (PC) 2, which explains 22.3% of the variation in the data, whereas all the bacterial taxa, except *E. coli*, span out on the PC1, explaining 29.3% of the variation in the data.

**Figure 3 animals-12-02519-f003:**
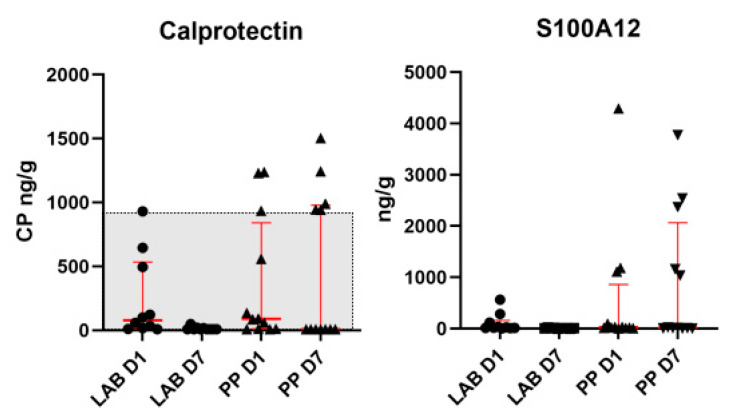
The plot shows the interindividual variation of fecal calprotectin (CP) concentrations and S100A12 in dogs given LAB or placebo on D1 and D7. Red lines show the medians and interquartile ranges, and the grey shaded area corresponds to the reference interval for CP (0–961 ng/g, Gastrointestinal Laboratory, Texas A&M University, TX, USA). Abbreviations: LAB, lactic acid bacteria; PP, placebo product.

**Figure 4 animals-12-02519-f004:**
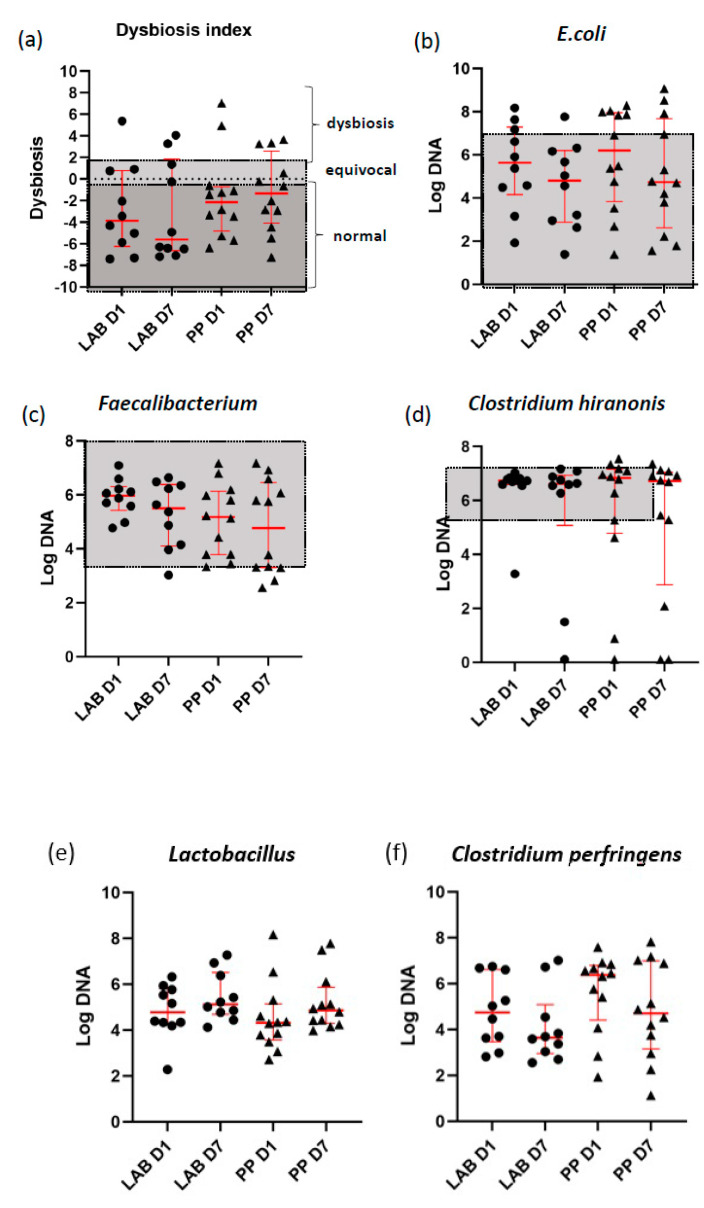
Dot plots showing (**a**) the canine fecal dysbiosis index (DI) and fecal abundances for (**b**) *E. coli*, (**c**) *Faecalibacterium*, (**d**) *Clostridium hiranonis*, (**e**) *Lactobacillus*, and (**f**) *Clostridium perfringens* for both groups of dogs on D1 and D7. Medians and interquartile ranges are indicated by red lines and the grey shaded areas correspond to the respective reference intervals. Abbreviations: LAB, lactic acid bacteria; PP, placebo product.

**Table 1 animals-12-02519-t001:** Overview of demographic factors, treatments, and occurrence of diarrhea in dogs included in the study.

Test Product	Breed	Age (Years)	Sex	Anesthesia	Reason for NSAID Treatment	Name of NSAID Treatment *	Occurrence of Diarrhea	Discontinued Treatment
Placebo	Finnish Lapphund	3	F	yes	Removal of benign skin tumor	Robenacoxib	yes	no
Placebo	German Short-haired Pointer	3	F	yes	Removal of benign skin tumor	Robenacoxib	no	no
Placebo	Pug	6	M	yes	Dental procedure	Robenacoxib	no	no
Placebo	Cavalier King Charles Spaniel	4	M	no	Benign prostate hypertrophy	Robenacoxib	yes	no
Placebo	Mixed breed	UN	F	yes	Mastectomy	Meloxicam	yes	yes (day 4)
Placebo	Pointer dog	2	F	yes	Pyometra surgery	Meloxicam	no	no
Placebo	Miniature Dachshund	10	F	no	Osteoarthritis	Meloxicam	no	no
Placebo	Dachshund	8	F	yes	Hemilaminectomy	UN	no	no
Placebo	Mixed breed	10	F	yes	Mastectomy	Robenacoxib	no	no
Placebo	Cocker Spaniel	1	F	yes	Patella luxation surgery	Robenacoxib	yes	yes (day 3)
Placebo	Jack Russel Terrier	8	F	yes	Hemilaminectomy	Robenacoxib	no	no
Placebo	Danish–Swedish Farmdog	10	F	yes	TPLO surgery	Robenacoxib	no	no
LAB	Alaskan Malamute	2	M	yes	Dental procedure	Robenacoxib	no	no
LAB	Cocker Spaniel	2	M	yes	Castration	Robenacoxib	no	no
LAB	Shih Tzu	4	F	yes	Pyometra surgery	Meloxicam	no	no
LAB	Finish Lapphund	10	M	yes	Removal of benign skin tumor	UN	no	no
LAB	Pomeranian	0.3	M	yes	Bone fracture surgery	Meloxicam	no	no
LAB	English Setter	10	F	yes	Removal of benign skin tumor	Robenacoxib	no	no
LAB	English Bulldog	7	F	yes	Pyometra surgery	Robenacoxib	yes	yes (day 3)
LAB	Alaskan Husky	14	F	no	Osteoarthritis	Robenacoxib	no	no
LAB	Medium Poodle	6	M	yes	Cystotomy due to urolithiasis	Robenacoxib	no	no
LAB	Drentsche Patrijshond	5	F	no	Diffuse pain related to the skeleton	Robenacoxib	no	no

Abbreviations: UN, not recorded; LAB, lactic acid bacteria treatment; TPLO, tibial plateau leveling osteotomy. * The doses were calculated based on the individual dog’s body weight and given enterally once daily. Dogs undergoing surgery were treated with parenteral NSAIDs for the first 24–48 h.

## Data Availability

All data are included in the manuscript.

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
