# Peer review of "Effects of Canine-Obtained Lactic-Acid Bacteria on the Fecal Microbiota and Inflammatory Markers in Dogs Receiving Non-Steroidal Anti-Inflammatory Treatment"

_animals, 2022, doi:10.3390/ani12192519_

Round 1

Reviewer 1 Report

Dears Authors,

Congratulations on the manuscript. Although there is a few limitations on the study all of them were reported in the text which increases the merit of this manuscript. Clinical studies are hard to make due to several factors that we can not control. 

The conclusions are compatible to the results found.

In the "Simple Summary" I do recommend remove most of the shortened word replacing by the complete word as such canine-obtained acid lactic bacteria, fecal calprotectin,  fecal calgranulin C instead of LAB, CP, and S100A12, respectively.

I suggest to treat S100A12 as calgranulin C and be consitent throw out of the abstract and text.

Line 95 - Were dogs eaten normally? During fecal sample collection they were eaten normally or some of the dogs had signal of anerexia? Please, provide comments

Line 111, 240, 264, - atempt to cientific names writting in italic and all throw the manuscript and legends.

Line 112 -  I do recommend in a next trial do not use cellulose as a placebo. Cellulose is minimally digestible (non digestible fiber) but have impact on stool quality. A tea spon of cellulose (˜2 g) for small dogs could have some impact. Do you think it could had some influence on your results?

Table 1 was helpfull to recognise the group of dogs. I was missing the body weight but by the breed we can figure out.

Conclusions are compatible to the obtained results.

Best regards,

Author Response

Please find the response attached.

Reviewer 2 Report

In this study, Herstad et al. aimed to test whether a probiotic product containing canine-adapted Lactobacillus strains would prevent NSAIDs-associated enteropathy. The authors' hypothesis is novel in the canine field and clinically important. The study suffers from patients’ variability and lack of statistical power, but the authors acknowledged this limitation. Their observation is still worth publishing after clarifying the points below:

·         The authors state the use of this probiotic product is "safe" without directly testing the safety (lines 29, 48, 317). Occurrences of diarrhea and vomiting are reported in the LAB group, which may or may not have been caused by the product. Please discuss the safety of this product in the Discussion, citing available safety data.

·         Please demonstrate that the incidence of the clinical adverse effects of NSAIDs (vomiting and diarrhea) in the placebo group was similar to what has been reported or observed in the authors' practice by citing literature or using historical data.

·         The rationale for the use of fecal CP/S100A12 to detect NSAIDs-induced enteropathy is well discussed in the Discussion, but it is unclear whether the data generated by the authors support it.

·         Is there a significant increase in fecal CP/S100A12 when D1 and D7 are compared in the placebo group?

·         Can the graphs in Figure 3 be changed to line graphs, connecting D1 and D7 of the same animals, so that the trends are clearer to the readers?

·         Can the fold changes (D7/D1) be compared to overcome the baseline variability?

· Table 1 shows two patients with 'UN' written for all columns. This is inappropriate, and the information should be available from the signed consent.

·         Method section is missing DNA extraction and qPCR details. Are the qPCR results normalized to anything?

Minor points

·         Line 114 'teaspoon': Please replace to, or add, a globally recognized and scientific unit (i.e. grams)

·         Please mention if any of the dogs received PPI.

Author Response

Please find the response attached.

Reviewer 3 Report

Dear Authors,

Thank you for submitting this interesting study that investigates that effect of LAB for dogs which are on NSAID treatment. This is an interesting and valuable study that provides some initial results in this area.

There are currently some revisions required. I have provided specific feedback on the PDF but please also see the following key points:

1. Case study. This study ultimately has a small sample size with a large amount of between-subject variation. As such, there is a limited ability to extend the results out to all animals. Please make sure this is considered carefully in the work. Similarly, consider in more detail what the future steps would be to analyse the effects of LAB more comprehensively.

2. Quality of data. I am a little concerned by the quality of data for the placebo individuals as there are some animals where the breed, sex and age are unavailable. If so little information has been collected, are the data actually reliable?

3. References. Currently there are inconsistencies in formatting to the Animals MDPI style. Please revise the references as per the author guidelines. 

Author Response

Please find the response attached.
